# Systematic morphological profiling of human gene and allele function via Cell Painting

**Mohammad Hossein Rohban[1], Shantanu Singh[1], Xiaoyun Wu[1], Julia B Berthet[2], Mark-Anthony Bray[1†], Yashaswi Shrestha[1], Xaralabos Varelas[2], Jesse S Boehm[1], Anne E Carpenter[1]***

[1]Broad Institute of MIT and Harvard, Cambridge, United States; [2]Department of Biochemistry, Boston University School of Medicine, Boston, United States

**Abstract** We hypothesized that human genes and disease-associated alleles might be systematically functionally annotated using morphological profiling of cDNA constructs, via a microscopy-based Cell Painting assay. Indeed, 50% of the 220 tested genes yielded detectable morphological profiles, which grouped into biologically meaningful gene clusters consistent with known functional annotation (e.g., the RAS-RAF-MEK-ERK cascade). We used novel subpopulation-based visualization methods to interpret the morphological changes for specific clusters. This unbiased morphologic map of gene function revealed TRAF2/c-REL negative regulation of YAP1/WWTR1-responsive pathways. We confirmed this discovery of functional connectivity between the NF-κB pathway and Hippo pathway effectors at the transcriptional level, thereby expanding knowledge of these two signaling pathways that critically regulate tumor initiation and progression. We make the images and raw data publicly available, providing an initial morphological map of major biological pathways for future study.

***For correspondence:** anne@broadinstitute.org

**Present address:** †Novartis Institutes for BioMedical Research, Cambridge, United States

**Competing interests:** The authors declare that no competing interests exist.

## Introduction

The dramatic increase in human genome sequence data has created a significant bottleneck. The number of genes and variants known to be associated with most human diseases has increased dramatically (*Amberger et al., 2015*). Unfortunately, the next step - understanding the function of each gene and the mechanism of each allele in the disease - typically remains non-systematic and labor-intensive. Most commonly, researchers painstakingly design, develop, and apply a disease-specific or biological process-specific assay.

Over 30% of genes in the human genome are of unknown function (*Leonetti et al., 2016*) and even annotated genes have additional functions yet to be uncovered. Furthermore, even when a gene's normal functions are known, methods are lacking to predict the functional impact of the millions of genetic variants found in patients. These gaps must be filled in order to convert the promise of human genome sequence data into clinical treatments.

Therefore, there is a widespread need for systematic approaches to functionally annotate genes and variants therein, regardless of the biological process or disease of interest. One general approach depends on guilt-by-association, linking unannotated genes to annotated ones based on properties such as protein-protein interaction data, sequence similarity, or, most convincingly, functional similarity (*Shehu et al., 2016*). In the latter category are profiling techniques, where dozens to hundreds of measurements are made for each gene perturbation and the resulting profile is compared against profiles for annotated genes. Various data sources can be used for profiling; gene expression is one that can be performed in relatively high-throughput and it has been proven useful

**eLife digest** Many human diseases are caused by particular changes, called mutations, in patients' DNA. A genome is the complete DNA set of an organism, which contains all the information to build the body and keep it working. This information is stored as a code made up of four chemicals called bases. Humans have about 30,000 genes built from DNA, which contain specific sequences of bases. Genome sequencing can determine the exact order of these bases, and has revealed a long list of mutations in genes that could cause particular diseases. However, over 30% of genes in the human body do not have a known role. Genes can serve multiple roles, some of which are not yet discovered, and even when a gene's purpose is known, the impact of each particular mutation in a given gene is largely uncatalogued. Therefore, new methods need to be developed to identify the biological roles of both normal and abnormal gene sequences.

For hundreds of years, biologists have used microscopy to study how living cells work. Rohban et al. have now asked whether modern software that extracts data from microscopy images could create a fingerprint-like profile of a cell that would reflect how its genes affect its role and appearance. While some genes do not necessarily carry a code with instructions of what a cell should look like, they can indirectly modify the structure of the cell. The resulting changes in the shape of the cell can then be captured in images. The idea was that two cells with matching profiles would indicate that their combinations of genes had matching biological roles too.

Rohban et al. tested their approach with human cells grown in the laboratory. In each sample of cells, they 'turned on' one of a few hundred relatively well-known human genes, some of which were known to have similar roles. The cells were then stained via a technique called 'Cell Painting' to reveal eight specific components of each cell, including its DNA and its surface membrane. The stained cells were imaged under a microscope and the resulting microscopy images analyzed to create a profile of each type of cell. Rohban et al. confirmed that turning on genes known to perform similar biological roles lead to similar-looking cells. The analysis also revealed a previously unknown interaction between two major pathways in the cell that control how cancer starts and develops.

In the future, this approach could predict the biological roles of less-understood genes by looking for profiles that match those of well-known genes. Applying this strategy to every human gene, and mutations in genes that are linked to diseases, could help to answer many mysteries about how genes build the human body and keep it working.

in predicting gene function (*Lamb et al., 2006*). In fact, high-throughput mRNA profiles were recently used to cluster alleles found in lung adenocarcinoma based on their functional impact, a precursor to therapeutic strategy for variants of previously unknown significance (*Berger et al., 2016*).

Images are a less mature data source for profiling but show tremendous promise. Morphological profiling data is complementary to transcriptional profiling data (*Wawer et al., 2014*) and is less expensive. Morphological profiling has succeeded across several applications, including grouping small-molecule perturbations based on their mechanism of action (*Caicedo et al., 2016*; *Bougen-Zhukov et al., 2017*), and grouping genes based on morphological profiles derived from cells perturbed by RNA interference (RNAi) (*Mukherji et al., 2006*; *Boutros and Ahringer, 2008*; *Fuchs et al., 2010*; *Pau et al., 2013*). One limitation of RNAi for morphological profiling is that the number of measurements must be limited or else the resulting profiles are dominated by off-target effects, especially seed effects (*Singh et al., 2015*). Some computational solutions have shown some promise in overcoming this problem for gene expression profiling (*Schmich et al., 2015*), but their utility is unproven for image-based profiling, and regardless RNAi does not permit analysis of gene variants, only knockdown. Modification of genes via CRISPR will require new libraries of reagents and is as yet untested in morphological profiling.

In the proof-of-concept work presented here, we tested morphological profiling using overexpression in human cells as a general approach to annotate gene and allele function. We profiled a reference series of well-known genes, and a small number of variants thereof, by Cell Painting. In

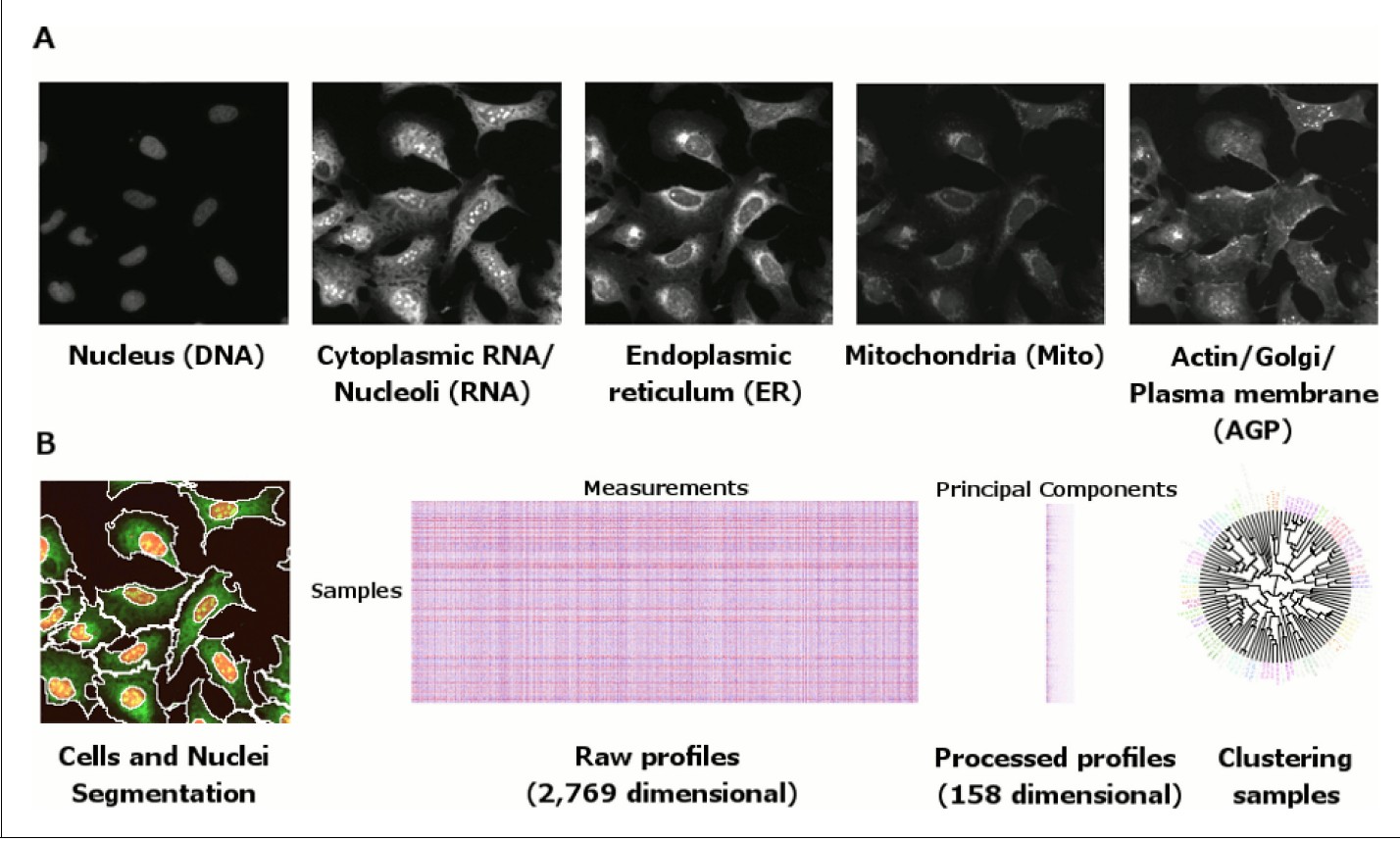

**Figure 1.** Morphological profiling by Cell Painting. (**A**) Example Cell Painting images from each of the five channels for a negative control sample (no gene introduced). (**B**) From left to right: Cell and nucleus outlines found by segmentation in CellProfiler; raw profiles (2769 dimensional) containing median and median absolute deviation of each of 1384 measurements over all the cells in a sample, plus cell count; processed profiles which are made less redundant by feature selection and Principal Component Analysis; dendrogram constructed based on the processed profiles (see *Figure 3*). Replicates are merged to produce a profile for each gene which is then compared against others in the experiment to look for similarities and differences.

particular, we wondered whether the information content of this strategy would outweigh potential limitations (e.g., due to cellular context or expression level). We found that the approach successfully clustered genes and alleles based on functional similarity, revealed specific morphological changes even when present in only a subpopulation of heterogeneous cells, and uncovered novel functional connections between important biological pathways.

## Results

### Morphological profiles from Cell Painting of expression constructs are sensitive and reproducible

To profile each exogenously expressed gene (or allele therein), we used our previously developed image-based profiling assay, called Cell Painting (*Gustafsdottir et al., 2013*; *Bray et al., 2016*). This microscopy-based assay consists of six stains imaged in five channels and revealing eight cellular components: DNA, mitochondria, endoplasmic reticulum, Golgi, cytoplasmic RNA, nucleoli, actin, and plasma membrane (*Figure 1A*). In five replicates in 384-well plate format, we infected U-2 OS cells (human bone osteosarcoma cells), chosen for their flat morphology and previous validation in the assay, with an arrayed 'reference' expression library of 323 open reading frame (ORF) constructs of partially characterized functions (*Supplementary file 1A*), a subset of which have been previously described (*Kim et al., 2016*). Of these, we prioritized analysis of the 220 constructs that were most

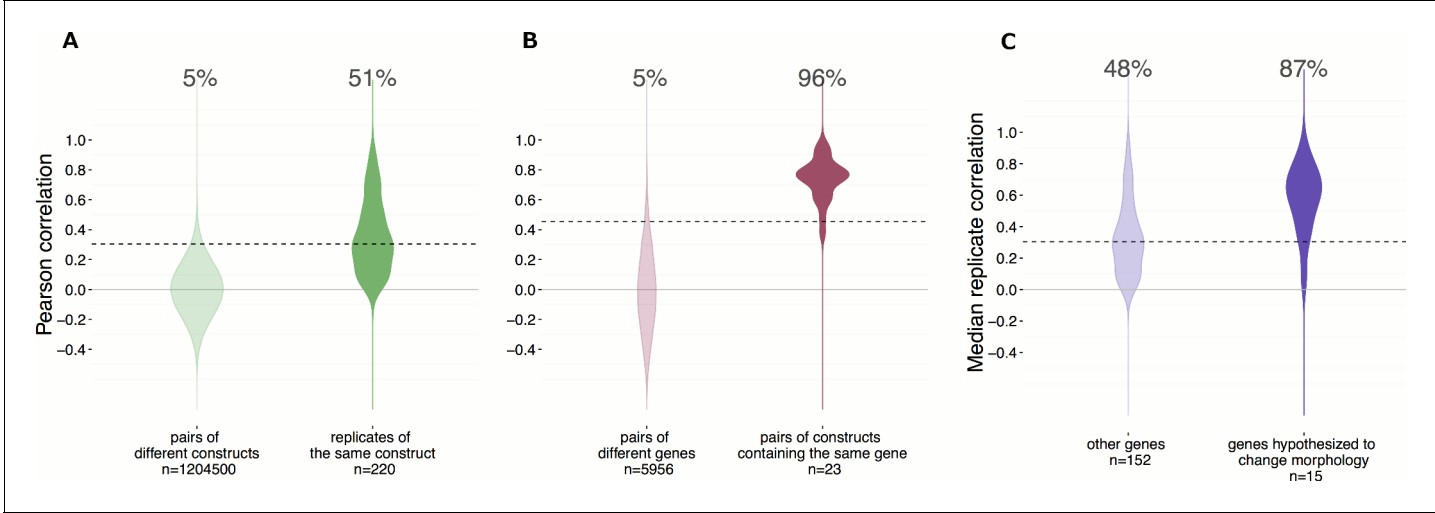

**Figure 2.** Morphological profiles are sensitive and reproducible, and show expected relationships. (**A**) 50% of the gene overexpression constructs produced a detectable phenotype by image-based profiling.Constructs yielding a reproducible phenotype ought to have a median correlation among replicates that is higher than the 95th percentile of correlations seen for pairs of different constructs; this is true for 51% (112 out of 220) of the constructs (as shown). Additionally, we removed two constructs that passed that filter but whose profiles were highly similar to negative control profiles (not shown), leaving 110 constructs (50%) for further analysis. (**B**) Of wild-type ORF pairs that both yielded a distinguishable phenotype, 96% showed significant correlation to each other. Correlations between the 23 pairs of constructs that are clones of the same gene (although with potential sequence variation or possibly different isoforms) were almost always much higher than correlations between pairs of constructs related to different genes. The threshold, shown as the dashed line, is set to 95th percentile of profile correlation for pairs of different genes. Profile correlation of these 23 pairs lie above the threshold. (**C**) Genes in pathways thought to regulate morphology were more likely to yield detectable phenotypes vs. the remainder of genes in the experiment. The same cutoff as in (**A**) is used to identify percentage of genes with a detectable phenotype. This percentage is 87% for the genes hypothesized to change morphology, while it is 48% for the other genes.

The following figure supplements are available for figure 2:

**Figure supplement 1.** Position artifacts do not contribute to the hit rate seen in the experiment.

**Figure supplement 2.** Strength of morphological phenotypes, according to annotated pathway.

closely representative of the annotated full length transcripts (see Materials and methods). Morphological profiles were extracted using CellProfiler for image processing, yielding 1384 morphological features per cell, and Python/R scripts for data processing, including feature selection and dimensionality reduction (*Figure 1B*, and see Materials and methods). This computational pipeline yielded a 158-dimensional profile for each of 5 replicates for each gene or allele tested.

Not all genes are likely to impact cellular morphology given the limitation of our experiment; using a single cell line at a single time point under a single set of conditions and stained with six fluorescent labels. We therefore first asked what fraction of these ORFs impacted morphology. Surprisingly, we found that 50% (110/220) of these ORF constructs induced reproducible morphological profiles distinct from negative control profiles (*Figure 2A*, and see Materials and methods). Next, we ruled out the possibility that position artifacts may have artificially inflated this result by taking an alternative pessimistic null distribution which takes well position into account (*Figure 2—figure supplement 1*). Therefore, we conclude that a single 'generic' morphological profiling assay can detect signal from a substantial proportion of genes in our reference set. We next turned to testing whether those signals are biologically meaningful and can lead to novel, unbiased discoveries about gene function.

## Morphological profiling is robust, showing expected relationships

Given that technical replicates produce similar morphological profiles, we next evaluated whether similarities between profiles induced by different constructs are meaningful. We began with the simplest case: for a subset of genes in the experiment, a 'wild-type' sequence (see

Materials and methods for important definitions) was captured in more than one ORF construct (23 pairs). These pairs either correspond to different physical cloning events and preparations but with highly similar full-length sequence (as defined in Methods; category a: nine pairs), or a substantive difference in their nucleotide sequence, for example, isoforms (category b: 14 pairs). We found that, as expected, the phenotypes of over-expressed wild-type ORFs of the same gene were more similar to each other, on average, than to randomly selected genes. Of the 23 pairs for which both wild-type ORFs yielded a phenotype distinguishable from negative controls, 22 (~96%) of the pairs' profiles were correlated more than expected by chance (*Figure 2B*, the one pair not meeting that threshold was in category b), confirming that different constructs with biological similarity indeed produce similar morphological profiles.

This result also confirms that the sequence differences seen in separately cloned wild-type constructs do not generally have a major functional impact, but we caution that any individual construct of interest may have an impactful mutation; thus the raw sequence data should be examined and testing alternate constructs for a gene may be recommended. Note that if, for example, only 50% of wild-type pairs showed high profile correlation, it would remain ambiguous whether it was caused by poor assay quality or by constructs' sequence mismatches. But in this particular case the mentioned near perfect consistency rules out either of the two possibilities. We also note that the 23 pairs analyzed here are located in different well locations on each plate; this result therefore also rules out widespread artifacts, such as plate position effects or metadata errors.

We suspected that the small number of engineered constitutively activating alleles for certain genes would, on average, yield a stronger phenotype than their wild-type counterparts. We indeed found that correlations between replicates of the constitutively activating allele were typically higher than correlations between replicates of the wild-type version of a gene (*Supplementary file 1B*; p-value=0.012, one-sided paired t-test).

We hypothesized that genes in pathways known to affect cellular morphology (RAC1, KRAS, CDC42, RHOA, PAK1, and genes related to the Hippo pathway) would be more likely to yield a morphological phenotype distinguishable from negative controls than other genes in the analysis. Indeed, we found this to be true (Fisher's test p-value=$3.7 \times 10^{-3}$) (*Figure 2C*). Reassured by this validation, we were curious which pathways would be most and least likely to yield detectable morphological phenotypes, recognizing that 'pathways' are neither separate nor well-defined entities. We found genes manually annotated as being in the Hippo, Hedgehog, cytoskeletal reorganization, and Mitogen-activated protein kinases (MAPK) pathways were more likely to result in a phenotype, whereas genes annotated as belonging to the JAK/STAT, hypoxia, and BMP pathways were among the least likely to yield a phenotype under the conditions tested (*Figure 2—figure supplement 2* and *Supplementary file 1C*). Nevertheless, the majority of pathways could be interrogated by morphological profiling.

## Morphological signature similarity captures known gene-gene relationships

Given the caveats and limitations of overexpressing genes (see Discussion), we next tested whether image-based profiling of expression constructs could capture relationships among genes known to be functionally related. Because a reliable and complete map of all gene-gene connections is not available, we evaluated the accuracy of our results via two approaches.

First, we compared our data to protein-protein interaction data from BioGRID (*Stark et al., 2006*). This is imperfect ground truth for judging our predictions because two proteins might physically interact without producing the same morphological phenotype when overexpressed, and genes in the same pathway might regulate the same phenotype without any physical interaction. Nevertheless, we expect that the corresponding proteins of gene pairs with highest profile similarity are more likely than average to physically interact. Indeed, looking at wild-type versions of genes showing a detectable phenotype (the 73 genes represented in the 110 constructs), the ratio of verified gene connections among the top 5% correlated gene pairs (9%, 13 verified out of 143 possible combinations) is significantly higher than that of other gene pairs (5%, 128 verified out of 2485 possible; Fisher's test p-value=0.04; *Supplementary file 1D*).

Second, we manually annotated each gene for the pathway with which it is associated. This approach is based on expert opinion and thus imperfect knowledge of all genes' function; furthermore many pathways interrelate, and genes in the same pathway are not expected to have identical

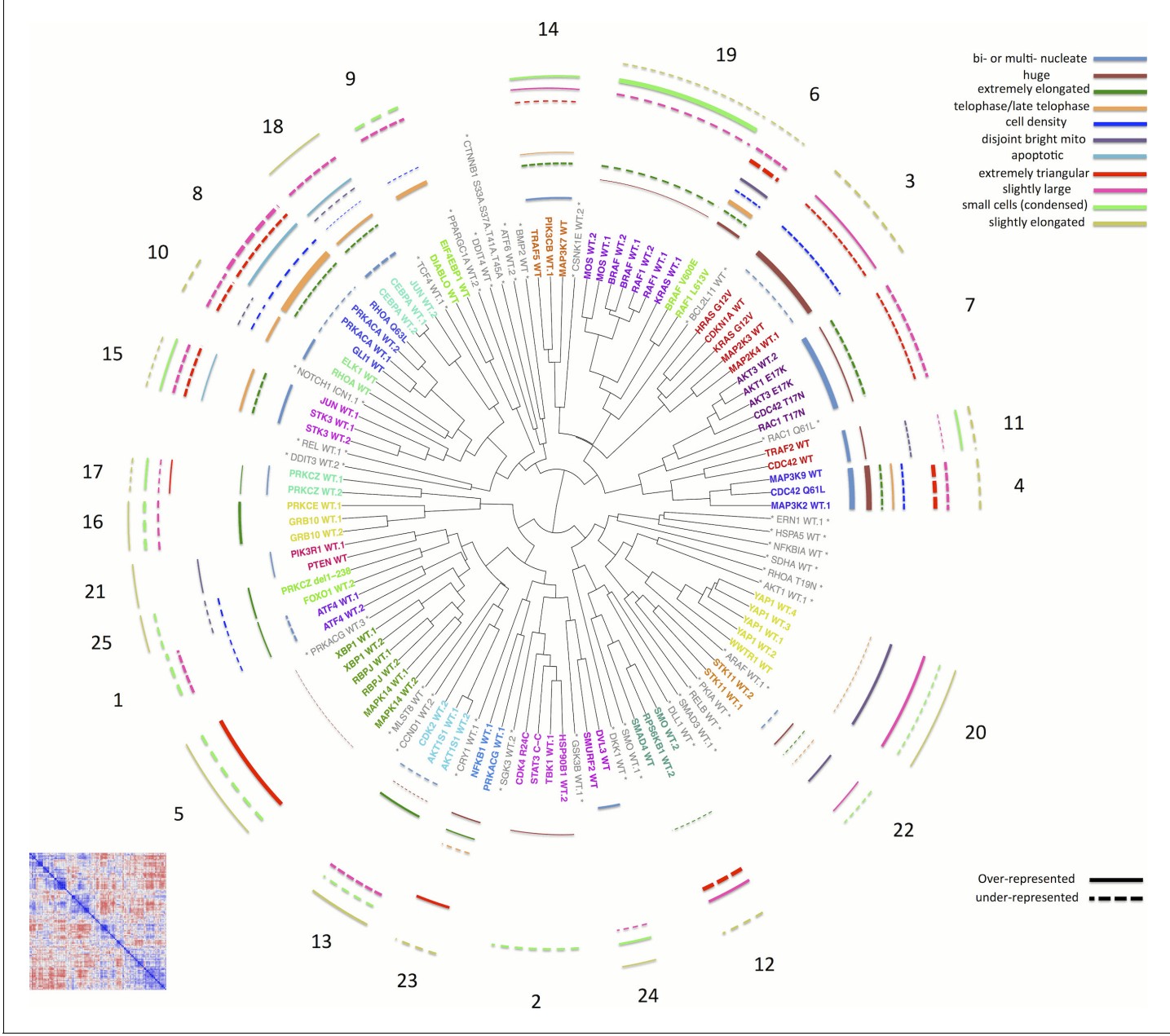

**Figure 3.** Morphological relationships among overexpressed genes/alleles, determined by Cell Painting. Correlations between pairs of genes/alleles were calculated and displayed in a correlation matrix (bottom left inset, full resolution is available as *Figure 3—figure supplement 1*). Only the 110 genes/alleles with a detectable morphological phenotype were included. The rows and columns are ordered based on a hierarchical clustering algorithm such that each blue submatrix on the diagonal shows a cluster of genes resulting in similar phenotypes. The correlations were then used to create a dendrogram (main panel) where the radius of the subtree containing a cluster shows the strength of correlation. The 25 clusters containing at least two constructs are printed on the dendrogram in arbitrary colored fonts, while gene names colored gray and marked by asterisks are those that do not correlate as strongly with their nearest neighbors (i.e., they are singletons or fall below the threshold used to cut the dendrogram for clustering). Each colored arc corresponds to a cell subpopulation as noted in the legend. Line thickness indicates the strength of enrichment of the subpopulation in the cluster samples compared to the negative control. Solid vs. dashed lines indicate the over- vs. under-representation of the corresponding subpopulation in a cluster, respectively. Note that the number next to each cluster in the dendrogram is referenced in the main text and corresponds to the numbered supplemental data file for each cluster.

The following figure supplements are available for figure 3:

**Figure supplement 1.** Correlation among the 110 genes/alleles with a detectable morphological phenotype.

*Figure 3 continued on next page*

*Figure 3 continued*

**Figure supplement 2.** Smoothed stability score across different cutoffs, in order to choose a threshold for cutting the dendrogram to form clusters.

**Figure supplement 3.** Common cell subpopulations seen across more than one cluster.

phenotypes given that their functions are rarely identical (most notably, overexpression of some may activate while others suppress a biological pathway or process). Nonetheless, we expect pairs of genes whose morphological profiles correlate highly to be more likely than average to be annotated in the same pathway vs. different pathways. Using the same 73 genes as in the previous analysis, the ratio of gene connections with the same-pathway annotation in the top 5% most-correlated gene pairs was 20% (29 pairs out of 143), significantly higher than the ratio for the remaining pairs (6%, 139 pairs out of 2485; Fisher's test p-value = $7.53 \times 10^{-9}$; *Supplementary file 1E*).

## An initial morphological map of gene function

Having quantitatively established that morphological profiling is sensitive, robust, and captures known gene-gene relationships, we explored these relationships in a correlation matrix (*Figure 3* bottom left and *Figure 3—figure supplement 1*). The overall structure, with multiple groupings along the diagonal, is consistent with the fact that the 110 constructs (73 unique genes) that showed a phenotype had been annotated as representing 19 different pathways. That is, we did not see large, homogeneous clusters, as would be expected if morphological profiling was sensitive to per-turbation but not highly specific. This rules out uniform toxicity induced by a large number of genes, for example. Neither did we see only signal along the diagonal, which would have indicated no strong similarity between any gene pairs.

We next created a dendrogram (*Figure 3*) and defined 25 clusters (see Materials and methods and *Figure 3—figure supplement 2*) to explore the similarities among genes. Pairs of wild-type ORFs almost always clustered adjacently, consistent with our quantitative analysis described above (*Figure 2B*). After retaining only one copy of replicate ORFs, we found that the majority of clusters (19 out of the 22 clusters containing more than one gene) were enriched for one or more Gene Ontology terms (*Supplementary file 1F*), indicating shared biological functions within each cluster.

Using this dendrogram, we began by interrogating three clusters that conformed well to prior biological knowledge. First, we analyzed Cluster 20, containing the two canonical Hippo pathway members YAP1 and WWTR1 (more detail in *Supplementary file 2* [PDFs A2–A20 and B2–B20 ] , and in a later section of the text). Both are known to encode core transcriptional effectors of the Hippo pathway (*Johnson and Halder, 2014*), and a negative regulator of these proteins, STK3 (also known as MST2), is the strongest anti-correlating gene for the cluster (*Supplementary file 2* [PDF A20], panel c1).

Second, we noted Cluster 21 is comprised of the two phosphatidylinositol 3-kinase signaling/Akt (PI3K) regulating genes, PIK3R1 and PTEN, both frequently mutated across 12 cancer types in The Cancer Genome Atlas (TCGA) (*Kandoth et al., 2013*). These results are consistent with previous observations that certain isoforms of PIK3R1 reduce levels of activated Akt, a dominant negative effect (*Abell et al., 2005*). AKT3 is in a cluster anti-correlated to the Cluster 21 ((Supplementary file 2 [PDF A21, panel b1]).

Third, we examined three clusters (19, 6 and 3) that included many MAPK-related genes. Cluster 19 is the largest example of a tight cluster of genes already known to be associated; it includes four activators in the RAS-RAF-MEK-ERK cascade: KRAS, RAF1 (CRAF), BRAF, and MOS. Notably, two constitutively active alleles of these genes, BRAF[V600E] (*Davies et al., 2002*) and RAF1[L613V] (*Wu et al., 2011*), form a separate cluster (Cluster 6) adjacent to their wild-type counterparts. Fur-thermore, the constitutively active RAS alleles HRAS[G12V] and KRAS[G12V] (*McCoy et al., 1984*) are in the next-closest cluster (Cluster 3), which also contains MAP2K4 and MAP2K3 (known to be acti-vated by Ras [*Shin et al., 2005*]), as well as CDKN1A (*Jalili et al., 2012*). By contrast, MAPKs that are known to be unrelated to the RAS-RAF-MEK-ERK cascade, such as MAPK14 in Cluster 5, are far away in the dendrogram.

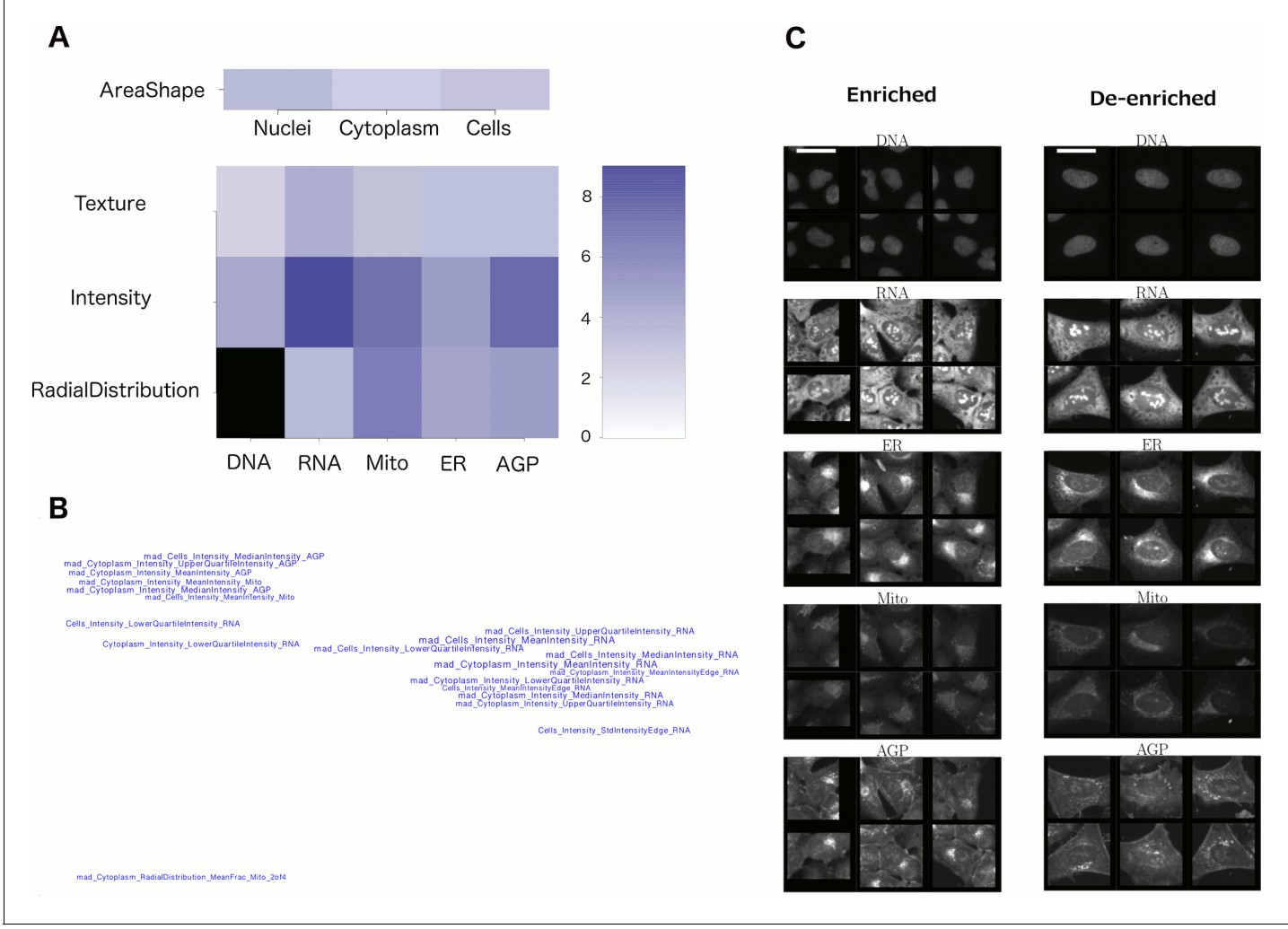

**Figure 4.** Visualizations used to interpret morphology of Cluster 19 (for other clusters, see *Supplementary file 2 [PDFs 1A–25A]*). (**A**) Feature Grid. RNA and AGP (actin, Golgi, plasma membrane) intensity contribute most to distinguishing the genes in Cluster 19 (KRAS, RAF1, BRAF, and MOS). Dark blue colors indicate higher median z-score of the relevant measurements for genes in the cluster relative to negative controls. As 'RadialDistribution' features do not exist for the DNA channel, it is colored in black. (**B**) Feature Map. The feature names showing the greatest difference between the cluster and negative controls are shown, based on largest absolute value of z-scores (full resolution version is available in Cluster 19A PDF). They are mapped in 2D space such that features that are highly correlated with each other across all genes' profiles are placed close together and thus can be interpreted together. Blue/red colored names indicate positive/negative sign of the z-score (i.e., blue indicates that the cluster shows higher values than controls). According to this map, the average intensity of AGP, RNA and Mito shows high variation for cells within samples in Cluster 19 (e.g., large mad_Cytoplasm_Intensity_MeanIntensity_AGP, where the prefix 'mad' refers to median absolute deviation, a robust form of standard deviation). (**C**) Sample images of a subpopulation of cells enriched and de-enriched for all genes in Cluster 19. Cells with asymmetric organelle distribution are highly over-represented for genes in the cluster, and cells with more even distribution of organelles are less abundant. Note that the exemplar cells are shown at the center of the patches. This explains the duplications observed in some patches. Scale bars are 39.36 $\mu m$ long. Pixel intensities are multiplied by five for display.

Overall, these results support the notion that connections between genes can be efficiently discovered using our approach.

## Visualization approaches to assist interpretation of morphological signatures

We hypothesized that the specific morphologic features that segregated each of the clusters would provide insight into gene function. Examining images (*Supplementary file 2 [PDF -19A*, panel 3]) or rank-ordered lists of features that distinguish individual profiles or clusters (*Supplementary file 1G*)

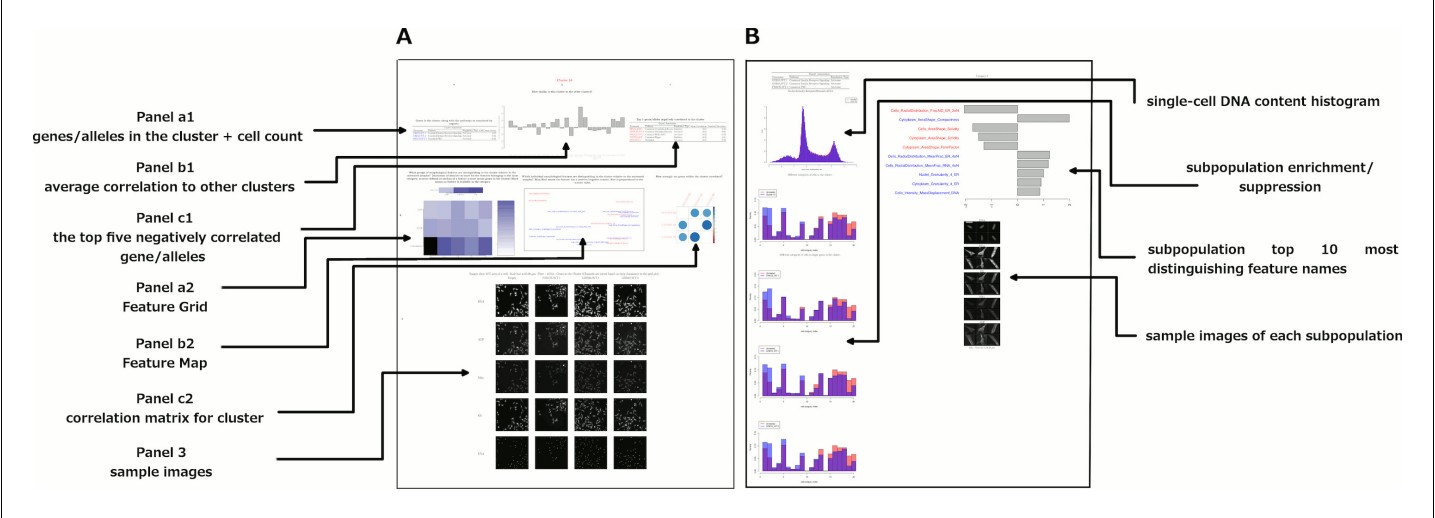

**Figure 5.** Data and visualizations supporting the morphological map for each cluster. For all 25 clusters, there are two corresponding Supplemental PDF files. Left: *Supplementary file 2* (type A PDFs, e.g., '1A.pdf') provide an overview of data about the cluster. Panel a1 lists the genes/alleles in the cluster as well as expert annotations regarding related pathways and the cell count (as a z-score) for each gene/allele. Panel b1 contains the average correlation of the cluster to other clusters, indicating uniqueness of the cluster's morphological phenotype. Panel c1 lists the top five negatively correlated gene/alleles to the cluster. Panel a2 shows the Feature Grid summarizing categories of morphological features distinguishing the cluster from the negative control. Panel b2 shows the Feature Map displaying the names of the top 20 morphological features distinguishing the cluster from the negative control, positioned based on similarity. Explanations for feature names can be found in the Methods section. Panel c2 shows a correlation matrix for just those genes/alleles in the cluster. Panel 3 contains sample images of fields of view of cells expressing each gene/allele in the cluster, along with images of the control for comparison. Right: *Supplementary file 2* (type B PDFs) contain multiple plots aiming to illustrate the phenotype based on single-cell data, including cell subpopulation enrichment/suppression in the cluster. First, a histogram of single-cell DNA content is shown for all cells from all genes/allele treatments in the cluster, indicating the overall cell cycle distribution. Next, bar plots show (for the cluster overall and for each gene in the cluster) which of 20 subpopulations of cells are enriched and suppressed relative to negative controls. Finally, each subsequent page of the PDF is devoted to the subpopulations whose representation differs from negative controls in a statistically significant way, whether enriched or suppressed (subpopulations which are very small in both the cluster and negative control samples are omitted). For each subpopulation, a bar plot shows the top 10 most-distinguishing feature names (versus negative control cells). Then, sample images are shown of individual representative cells from each subpopulation.

is tedious and lacks sensitivity for all but the most obvious of phenotypes, confirming that quantitative morphological profiling is more sensitive than the human visual system.

We therefore devised several strategies to enhance biological interpretability from these experiments and applied these in combination. First, we grouped features into meta-features based on their type of measurement, i.e., shape, texture, intensity, etc., and the cell constituents to which they are related, to create a Feature Grid (*Figure 4A*). Second, we performed unsupervised grouping of features by mapping the top 20 most-distinguishing features for each cluster onto a plane, creating a Feature Map (*Figure 4B*), in which highly correlated features are mapped nearby each other (see 'Feature Interpretation' in Methods for an explanation of individual feature names). In certain cases, these visualizations revealed the nature of the morphological phenotype (e.g., nuclear shape abnormalities *Supplementary file 2 [PDF 7A]*), but for others these approaches did not suffice to yield an obvious phenotypic conclusion (e.g., for Cluster 19, *Figure 4A and B*).

Third, we hypothesized that leveraging the single-cell resolution of image-based profiling might be highly sensitive in enhancing interpretation, particularly for cases where only a subset of cells is distinctive from negative controls. To test this, for each given cluster of genes together with negative controls we identified 20 subpopulations using k-means clustering on single cell data. We calculated the abundance of cells in each of the 20 subpopulations to determine which are over/under-represented relative to controls for the given cluster (corresponding images are shown; *Supplementary file 2* [PDFs 1B–25B]). For example, the MAPK pathway activators in Cluster 19 show increased prevalence of a subpopulation of cells with strongly asymmetric ER, mitochondria, and Golgi staining, indicating a cell polarization phenotype (*Figure 4C*, and *Supplementary file*

*2 [PDF 19B]*, Categories one and two), for which there is evidence in the literature (*Samaj et al., 2004*; *Elsum et al., 2013*; *Godde et al., 2014*). This phenotype was not captured by manual inspection nor the first two approaches (e.g., *Supplementary file 2 [PDF 19A*, panels a2 and b2]).

Encouraged by this, we supplemented the morphological map by compiling these and other visualizations into PDF files for each cluster, summarized in *Figure 5* and provided in full as *Supplementary file 2*. We also noticed that certain subpopulations were similar across several clusters (*Figure 3—figure supplement 3* shows sample cell images of each such subpopulation); we annotated their enrichment/de-enrichment on the dendrogram (*Figure 3*).

Using these visualizations, we began by interrogating three adjacent and correlating clusters (Clusters 4, 7, and 11) contain wild-type and mutant alleles of CDC42, a gene encoding a Rho family GTPase with diverse roles in cell polarity, morphology, and migration (*Melendez et al., 2011*; *Martin, 2015*). Cluster 4 contains the constitutively active mutant CDC42 Q61L (*Nobes and Hall, 1999*) as well as MAP3K2 and MAP3K9. The highly similar Cluster 7 contains the dominant negative alleles CDC42 T17N (*Nobes and Hall, 1999*) and RAC1 T17N (*Zhang et al., 1995*), a related RAS superfamily member. That activating and inhibiting alleles would yield similar phenotypes when overexpressed is not surprising for CDC42 (*Melendez et al., 2011*). Cluster 7 also contains isoforms and alleles of AKT: specifically, AKT3 and the constitutively active E17K alleles of both AKT1 and AKT3 (*Kim et al., 2008*; *Davies et al., 2008*). Akt is known to be essential for certain Cdc42-regulated functions (*Higuchi et al., 2001*) and vice versa (*Stengel and Zheng, 2012*). Finally, the nearby Cluster 11 (which is discussed in more detail later) contains the wild-type form of CDC42 as well as TRAF2, a canonical NF-κB activator; these two are known to interact and share functions in actin remodeling (*Marivin et al., 2014*). We also note that anti-correlating genes to these clusters (generally in Clusters 13 and 21) are consistent with existing knowledge, including (a) AKT family member AKT1S1 (a Proline rich AKT substrate, PRAS40 (*Kovacina et al., 2003*; *Wiza et al., 2014*), *Supplementary file 2 [PDF 7A*, panels b1 and c1]) (b) CDK2 (a known target of Akt [*Maddika et al., 2008*]), (c) PIK3R1 and PTEN in Cluster 21, described previously, which have known interactions with AKT (*Cheung and Mills, 2016*; *Hemmings and Restuccia, 2015*). Thus, all of these connections have previously been identified.

Subpopulation visualization revealed that Clusters 4, 7, and 11 are enriched in cells that are huge and binucleate (*Figure 3*, example images shown in *Supplementary file 2 [PDF 4B]*). Genes in all three clusters also show irregularities in DNA content, namely, an enrichment in cells with sub-2N DNA content, a decrease in cells with 2N DNA content, and, for most genes, a decrease in cells with S phase and 4N DNA content, indicating a significant amount of DNA fragmentation and thus apoptosis (DNA histograms in *Supplementary file 2 [PDFs 4B, 7B, and 11B]*). These phenotypes are consistent with these genes' known role in the cell cycle and cell polarity (*Chircop, 2014*).

As a second test case, we examined Cluster 8, which contains PRKACA (the catalytic subunit α of protein kinase A, PKA) and two of its known substrates: GLI1 (a transcription factor mediating Hedgehog signaling)(*Asaoka, 2012*), and RHOA$^{Q63L}$ (a Ras homolog gene family member) (*Lang et al., 1996*; *Rolli-Derkinderen et al., 2005*). The highly similar Cluster 10 contains the wild-type RHOA, as well as ELK1 which is also linked to the Rho GTPase family and PKA (*Bachmann et al., 2013*; *Murai and Treisman, 2002*).

We investigated the morphological changes causing these genes to cluster. RhoA is a known regulator of cell morphology and cell rounding is a known related phenotype (*Oishi et al., 2012*). We found that indeed all members of Clusters 8 and 10 significantly induce cell rounding (*Supplementary file 1H*). Although cell count is lower for genes Clusters 8 and 10, the degree varies greatly (from z-score −0.67 to −3.02, *Supplementary file 2 [PDFs 8A and 10A* , panel a1]), ruling out that simple sparseness of cells explains their high similarity in the assay. As well, the overall DNA content distribution of the cell populations appears relatively normal (*Supplementary file 2 [PDFs 8B and 10B]*). Subpopulation extraction provides a satisfying biological explanation for these clusters' distinctive phenotype: the increased roundness and strong variation in intensity levels (per the Feature Grid) across the population stems from an increased proportion of telophase, anaphase, and apoptotic cells (*Figure 3* and *Supplementary file 2 [PDFs 8B and 10B]*).

We therefore conclude that the morphological map can link related genes to each other and that the morphological data can provide insight into their functions, particularly with the help of subpopulation visualization.

## An unexpected relationship between the Hippo pathway and regulators of NF-κB signaling (Clusters 11, 20, and 22)

We wondered whether novel relationships might emerge from our unbiased classification of gene and allele function based on morphologic profiling. We noticed that the known regulator of NF-κB signaling, TRAF2 (in Cluster 11, together with CDC42) (*Grech et al., 2004*; *Tada et al., 2001*), yields a signature strongly anti-correlated to YAP1/WWTR1 (Cluster 20), which encode the transcriptional effectors of the Hippo pathway, YAP (Yes-associated protein) and TAZ (Transcriptional co-activator with a PDZ-domain). The Hippo pathway and NF-κB signaling are critical regulators of cell survival and differentiation, and dysregulation of these pathways is implicated in a number of cancers (*Varelas, 2014*; *Hoesel and Schmid, 2013*; *Tornatore et al., 2012*), but we found no evidence in the literature (in particular through BioGRID) of physical interaction between the proteins encoded by Cluster 11 genes and Cluster 20 genes. Confirming our approach, a functional connection between CDC42 (Cluster 11) and YAP1 (Cluster 20) has been identified: deletion of CDC42 phenocopies the loss of YAP1 in kidney-specific conditional knockouts in mice (*Reginensi et al., 2013*). Still, the NF-κB pathway (and in particular the Cluster 11 member TRAF2), has not been closely tied to YAP and TAZ in human cells (see Discussion).

We first wanted to characterize Clusters 11 and 20 to confirm that relationships within each cluster are supported in the literature. Indeed we found evidence for most of the within-cluster connections. CDC42 and TRAF2 (Cluster 11) physically interact and share functions in actin remodeling (*Marivin et al., 2014*). As described in a prior section YAP/TAZ (Cluster 20) are known to share functional similarities in the Hippo pathway, being regulated by, and also regulating, cytoskeletal dynamics. Consistent with these known functions, we found that a core effector of the Hippo pathway which functions to restrict YAP/TAZ nuclear activity, STK3 (which encodes the Mst2 kinase) (*Meng et al., 2016*), has a morphological signature strongly anti-correlated to YAP1/WWTR1 (*Supplementary file 2 [PDF 20A*, panel c1]). We note that although STK3 and TRAF2 are both moderately anti-correlated with YAP/TAZ (Cluster 20), STK3 and TRAF2 are not themselves highly correlated, indicating each has a different subset of phenotypes that anti-correlate to YAP/TAZ. We also note that two clones that express another regulator of YAP activity, STK11, form Cluster 22 which falls nearby YAP1/WWTR1; a connection between STK11 and YAP has been identified (albeit with opposite directionality, identified via knockdown of STK11 [*Mohseni et al., 2014*]). Further, YAP1 is among the highest anti-correlating genes to REL (data not shown; REL is a singleton in the dendrogram and thus not in a cluster), whose protein product, c-Rel, has a known connection to TRAF2 (*Jin et al., 2015*). These results reaffirm that the Cell Painting-based morphological signatures are a useful reporter of biologically meaningful connections among genes in these pathways.

Given the striking inverse correlation between YAP1/WWTR1 and TRAF2, we sought to confirm a negative regulatory relationship between the Hippo and NF-κB pathways by multiple orthogonal methods.

First, we explored the observed inverse morphological impact using the Cell Painting data. The morphological impact of genes in Cluster 11 and 20 is quite strong (median replicate correlation is at the 74th and 81st percentile, and average within-group correlations are 0.66 and 0.73). Subpopulation analysis showed that Cluster 20 (YAP1, WWTR1) is enriched for cells that are slightly large, slightly elongated, and have disjoint, bright mitochondria patterns, whereas Cluster 11 (TRAF2, CDC42) is de-enriched for those subpopulations and instead enriched for binucleate cells, very large cells, and small cells with asymmetric organelles (*Figures 3*, *6A and B*).

Second, given that YAP/TAZ are transcriptional regulators, we analyzed gene expression data. Using the same constructs as in our Cell Painting experiment, we found an anti-correlated relationship at the mRNA level, consistent with the anti-correlation we had seen in morphological space. To do this, we used Gene Set Enrichment Analysis (*Subramanian et al., 2005*) and publicly available data, which includes data from four to nine different cell lines at one to four time points (https://clue.io). Time point refers to the duration of treating the cells with over-expression constructs until the time gene expression readouts are made. This analysis revealed that the NF-κB pathway is the pathway most enriched among genes whose overexpression results in down-regulation of known YAP1 targets, CTGF, CYR61, and BIRC5 (*Zhao et al., 2008*) (Benjamini and Hochberg (BH) adjusted p-value = $2 \times 10^{-8}$ in *Supplementary file 1I*, and *Figure 6C*), with TRAF2 being among the genes contributing to this enrichment (*Supplementary file 1I*). We also saw enrichment of NF-κB pathway

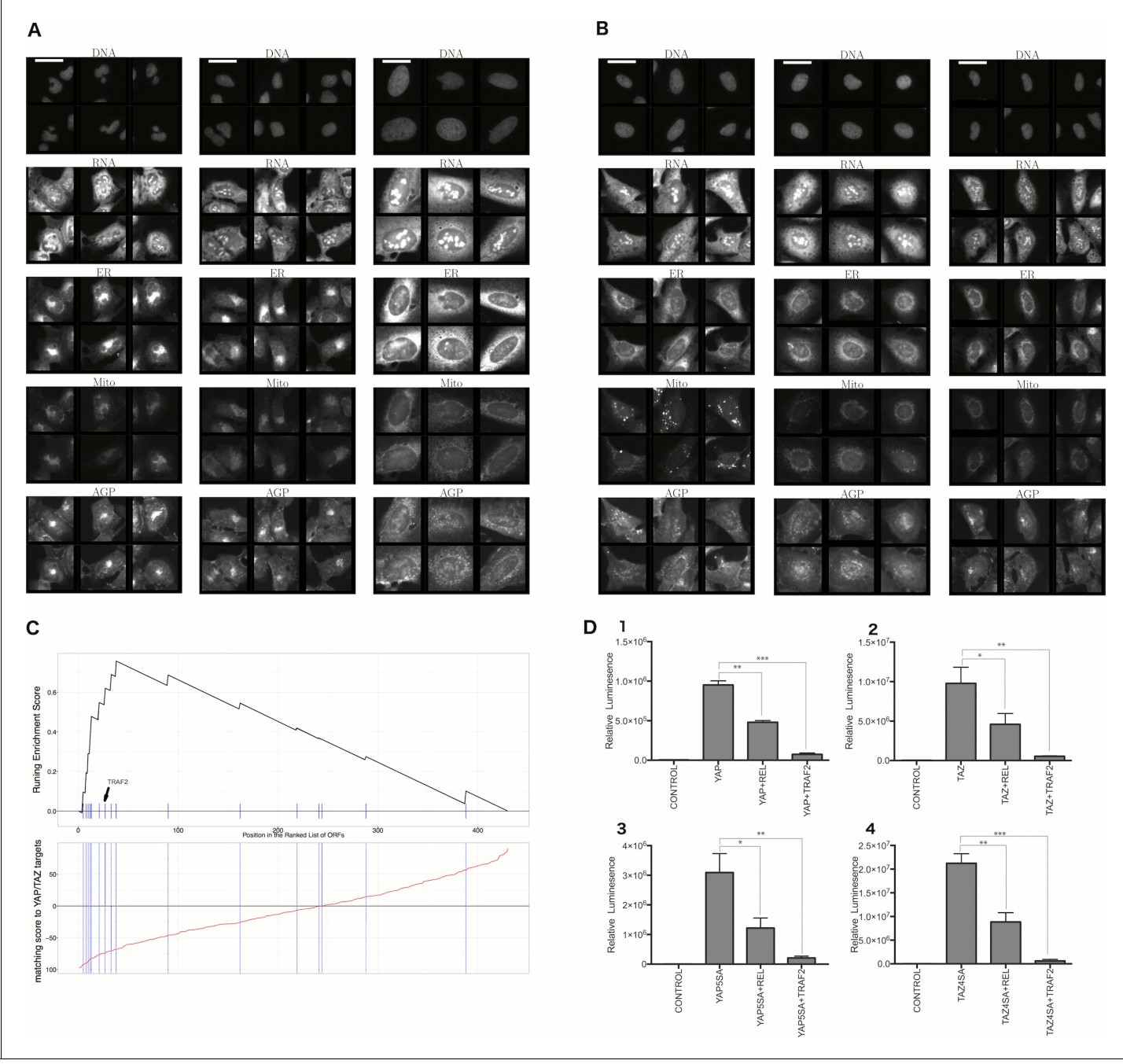

**Figure 6.** Morphological and transcriptional cross-talk between the Hippo pathway and regulators of NF-κB signaling. (**A**). The TRAF2/CDC42 cluster (Cluster 11) is enriched for bi-nucleate cells, small cells with asymmetric organelles, and huge cells. Note that exemplar images shown are not labeled as to the actual gene they are associated with. Rather they are only supposed to provide a visual insight of the cell morphologies which are enriched in the gene cluster. (**B**) The YAP1/WWTR1 cluster (Cluster 20) is enriched for cells with bright disjoint mitochondria patterns, slightly large cells, and slightly elongated cells. Scale bars are 39.36 $\mu m$ long. Pixel intensities are multiplied by five for display. (**C**) Gene Set Enrichment Analysis (GSEA) reveals that gene overexpression leading to down-regulation of YAP1 targets (CTGF, CYR61, and BIRC5) are enriched for regulators of the NF-κB pathway (Enrichment Score p-value = $8.19 \times 10^{-5}$). The horizontal axis gives the index of ORFs sorted based on the average amount of down-regulation of the YAP1 targets. Each blue hash mark on this axis indicates an NF-κB pathway member. The running enrichment score, which can range from −1 to 1, is plotted on the vertical axis and quantifies the accumulation of NF-κB pathways members on the sorted list of ORFs. (**D**) TRAF2 and REL suppress YAP and TAZ transcriptional activity. REL and TRAF2 suppress the ability of wild-type (**D1**) YAP and (**D2**) TAZ to drive the expression of a TEAD-regulated luciferase reporter. Activity of nuclear active mutants of (**D3**) YAP (5SA) and (**D4**) TAZ (4SA) are similarly suppressed. Luciferase reporter activity was measured in HEK293T cells co-transfected with expression constructs as indicated and a TEAD luciferase reporter was used to measure YAP-directed transcription. (* p-value<0.05, ** p-value=0.001, *** p-value<0.0001).

*Figure 6 continued on next page*

*Figure 6 continued*

The following figure supplements are available for figure 6:

**Figure supplement 1.** Gene Set Enrichment Analysis (GSEA) reveals that overexpression constructs sorted based on their similarity to YAP1/WWTR1 overexpression (in terms of impact on particular mRNA targets), are enriched for regulators of the NF-κB pathway (Enrichment Score p-value=0.0019).

**Figure supplement 2.** Gene Set Enrichment Analysis (GSEA) reveals that overexpression constructs sorted based on their similarity to TRAF2/REL overexpression (in terms of impact on particular mRNA targets), are weakly enriched for regulators of the Hippo pathway (Enrichment Score p-value=0.024).

members when testing a data-driven set of targets of YAP1/TAZ (*Figure 6—figure supplement 1*, see Materials and methods). In the inverse analysis, genes that alter the levels of TRAF2/REL common targets are weakly enriched in Hippo pathway members (*Figure 6—figure supplement 2*, see Materials and methods). This is consistent with the hypothesis that NF-κB members can downregulate YAP/TAZ targets but not strongly vice versa.

Finally, we more directly confirmed negative crosstalk between NF-κB effectors and YAP/TAZ using a synthetic TEAD luciferase reporter that is YAP/TAZ responsive (*Dupont et al., 2011*). Importantly, these confirmatory experiments used different cellular contexts and perturbation constructs versus the original Cell Painting data. Co-expression of the NF-κB pathway effectors TRAF2 or C-REL with YAP or TAZ led to significantly lower reporter activity than expression of YAP or TAZ alone (*Figure 6D1 and D2*). Intriguingly, mutants of YAP or TAZ that are insensitive to negative regulation by the Hippo pathway (YAP-5SA and TAZ-4SA; [*Zhao et al., 2008*]) remained sensitive to suppression of transcriptional activity by TRAF2 and C-REL, indicating that the negative relationship we identified may be independent of canonical upstream Hippo pathway signals (*Figure 6D3 and D4*).

## Discussion

We conclude that connections among genes can be profitably analyzed using morphological profiling of overexpressed genes via the Cell Painting assay. In a single inexpensive experiment, we were able to rediscover a remarkable number of known biological connections among the genes tested. Further, we found that morphological data from the Cell Painting assay, together with novel subpopulation visualization methods, can be used to flesh out the functionality of particular genes and/or clusters of interest.

By adopting a two-pronged approach, merging this Cell Painting morphological analysis with transcriptional data, we were able to identify an unexpected relationship in human cells between two major signaling pathways, Hippo and NF-κB, both under intense study recently for their involvement in cancer. Through validation of these clustered genes, we have identified that YAP/TAZ-directed transcription is negatively regulated by NF-κB pathway effectors and our data suggests a novel regulatory mechanism that is independent of upstream Hippo kinases.

To date, there has been little evidence of the intersection between these important signaling pathways. Recent work examining osteoclast-osteoblast differentiation has suggested that Hippo pathway kinases, such as Mst2, may affect the NF-κB pathway through phosphorylation of IkB proteins, thereby promoting nuclear translocation of NF-κB transcription factors (*Lee et al., 2015*). TAZ was found to be a direct target of NF-κB transcription factors and its expression is regulated via NF-κB signaling (*Cho et al., 2010*). Our work, however, supports a possible additional mode of interaction, whereby regulators of NF-κB signaling directly regulate the function of Yap and Taz as transcriptional co-factors. Recent work has demonstrated, in *Drosophila*, that NF-κB activation via Toll receptor signaling negatively regulates the transcriptional activity of Yorkie, the homolog of YAP/TAZ, through activation of canonical hippo pathway kinases (*Liu et al., 2016*). The work described here identifies, for the first time in a mammalian system, that a negative regulatory relationship exists between NF-κB activation and YAP/TAZ transcriptional function. Furthermore, we have identified that this regulation of YAP/TAZ occurs in a manner that is independent of Hippo pathway-

mediated phosphorylation events on YAP/TAZ, suggesting a more direct relationship between NF-κB and YAP/TAZ signaling.

In this work, we tested quantitatively and explored qualitatively the connections among genes revealed by morphological profiling. Our underlying hypothesis was that functionally similar genes would generally yield morphologically similar cells when overexpressed, and indeed we found this to be the case. Still, some discussion of this point is warranted. Most commonly, gene overexpression will result in activation of the corresponding pathway via amplification of the endogenous gene's function. However, it is important to note that the profiling strategy to discover functional relationships does not assume or require this. For example, overexpression could also disrupt a protein complex, producing a trans-dominant negative effect that results in precisely the opposite phenotypic effect (*Veitia, 2007*). In still other cases, overexpression of a particular gene may not affect any of the normal functions of the gene (producing a false negative signal), or trigger a stress response (yielding a confounded profile), or produce a complicated response, due to feedback loops. Further, artifactual phenotypes could be seen, e.g., if overexpression yields a non-physiological interaction among proteins or toxic aggregates. Nevertheless, despite these caveats and complications, our results indicate that valuable information could be gleaned from the similarity and dissimilarity of the morphological perturbations induced by gene overexpression. Using overexpression avoids the complications of RNAi off-target effects (often due to seed effects), which were far more prevalent (impacting 90% of constructs in our recent study [*Singh et al., 2015*]).

In addition to functionally annotating genes, as demonstrated here, one particularly appealing application enables personalized medicine: it should be feasible to use morphological profiling to predict the functional impact of various disease alleles, particularly rare variants of unknown significance. This has recently been successful using mRNA profiles (*Berger et al., 2016*). Thus, an even more exciting prospect would be to combine mRNA profiles with morphological profiles to better predict groups of alleles of similar mechanism, and ultimately to predict effective therapeutics for each group of corresponding patients.

We make all raw images, extracted cellular features, calculated profiles, and interpretive visualizations publicly available, providing an initial morphological map for several major signaling pathways, including several unexplored connections among genes for further study (see *Supplementary file 2*). Expanding this map to full genome scale could prove an enormously fruitful resource.

## Materials and methods

### cDNA constructs used for expression

The Reference Set of human cDNA clones utilized here has been previously described (*Kim et al., 2016*); ~90% of these constructs induce expression of the intended gene greater than two standard deviations above the control mean. Briefly, wild-type ORF constructs were obtained as Entry clones from the human ORFeome library version 8.1 (http://horfdb.dfci.harvard.edu) with additional templates generously provided by collaborating laboratories, and cloned into the pDONR223 Gateway Entry vector. In addition, here, to maximize coverage of cellular pathways, we included additional clones with minimal sequence deviations from the intended templates. Sanger sequencing of Entry clones verified the intended transcripts and, if applicable, the intended mutation. Entry constructs and associated sequencing data will be publicly available via www.addgene.org and may also be available via members of the ORFeome Collaboration (http://www.orfeomecollaboration.org/), including the Dana-Farber/Harvard Cancer Center (DF/HCC) DNA Resource Core DNA Repository (http://www.dfhcc.harvard.edu/core-facilities/dna- resource/) and the DNASU Plasmid Repository at ASU Biodesign Institute (http://dnasu.asu.edu/DNASU/Home.jsp). Clone requests must include the unique clone identifier numbers provided in the last column of *Supplementary file 1A* (e.g. ccsbBroadEn_12345 as an example for a specific entry clone and ccsbBroad304_12345 as an example for a specific expression clone). ORFs were transferred to the pLX304 lentiviral expression vector (*Yang et al., 2011*) by LR (attL x attR) recombination.

For simplicity, throughout this paper 'wild-type' refers to ORFs found in the original collection without a particular known mutation intentionally engineered. Due to natural human variation, and occasional cloning artifacts, there are often non-identical matches of such constructs to reference sequence; these differences are fully documented for each construct and sequence data will be

publicly available through AddGene, in addition to the sequencing data for the original Entry clones for the genome-scale library (*Yang et al., 2011*).

## Cell lines

U-2 OS cells (human bone osteosarcoma cells), RRID:CVCL_0042, were obtained from ATCC and propagated in the William Hahn lab; they were not additionally authenticated prior to this experiment. The cell line tested negative for mycoplasma prior to this experiment. HEK293T cells, RRID: CVCL_0063, were obtained from ATCC. The cell line was validated by STR profiling (Genetica DNA Laboratories) and was negative for mycoplasma as measured by MycoAlert Mycoplasma Detection Kit (Lonza, Walkersville, MD).

## Lentiviral transduction for morphological profiling

We followed our previously described protocol (*Kim et al., 2016*; *Berger et al., 2016*) except for durations of some steps. Briefly, cells were plated in 384-well plates and transduced with lentiviral particles carrying ORF constructs the next day. Viral particles were removed 18–24 hr post-infection and cells cultured for 48 hr until staining and imaging (72 hr total post-transduction). The experiment was conducted in five replicates, each in a different plate. The number of replicates being five was decided based on prior experiments (*Bray et al., 2016*).

## Cell staining and imaging

The Cell Painting assay followed our previously published protocol (*Bray et al., 2016*). Briefly, eight different cell components and organelles were stained with fluorescent dyes: nucleus (Hoechst 33342), endoplasmic reticulum (concanavalin A/AlexaFluor488 conjugate), nucleoli and cytoplasmic RNA (SYTO14 green fluorescent nucleic acid stain), Golgi apparatus and plasma membrane (wheat germ agglutinin/AlexaFluor594 conjugate, WGA), F-actin (phalloidin/AlexaFluor594 conjugate) and mitochondria (MitoTracker Deep Red). WGA and MitoTracker were added to living cells, with the remaining stains carried out after cell fixation with 3.2% formaldehyde. Images from five fluorescent channels were captured at 20x magnification on an ImageXpress Micro epifluorescent microscope (Molecular Devices): DAPI (387/447 nm), GFP (472/520 nm), Cy3 (531/593 nm), Texas Red (562/624 nm), Cy5 (628/692 nm). Nine sites per well were acquired, with laser based autofocus using the DAPI channel at the first site of each well.

## Image processing and feature extraction

The workflow for image processing and cellular feature extraction has been described elsewhere (*Bray et al., 2016*), but we describe it briefly here. CellProfiler (*Carpenter et al., 2006*) software version 2.1.0 was used to correct the image channels for uneven illumination, and identify, segment, and measure the cells. An image quality workflow (*Bray et al., 2012*) was applied to exclude saturated and/or out-of focus wells; six wells containing blurry images were excluded, retaining 1914 plate/well combinations in the experiment. Cellular morphological, intensity, textural and adjacency statistics were then measured for the cell, nuclei and cytoplasmic sub-compartments. The 1402 cellular features thus extracted were normalized as follows: For each feature, the median and median absolute deviation were calculated across all untreated cells within a plate; feature values for all the cells in the plate were then normalized by subtracting the median and dividing by the median absolute deviation (MAD) times 1.4826 (*Chung et al., 2008*). Features having MAD = 0 in any plate were excluded, retaining 1384 features in all. The image data along with the extracted morphological features at the per-cell level were made publicly available in the Image Data Repository under DOI 10.17867/10000105.

## Profiling and data preprocessing

The code repository for the profiling and all the subsequent analysis is publicly available at https://github.com/carpenterlab/2017_rohban_elife (*Carpenter, 2017*) (with a copy archived at https://github.com/elifesciences-publications/2016_rohban_submitted). We will next explain details of each analysis step implemented in the code. Single cell measurements in each well and plate position are summarized into the profiles by taking their median and median absolute deviation (abbreviated as 'MAD' or 'mad' in some tables) over all the cells. Although this method does not explicitly capture

population heterogeneity, no alternate method has yet been proven more effective (*Ljosa et al., 2013*). We also include the cell count in a sample as an additional feature. This results in a vector of 2769 elements describing the summarized morphology of cells in a sample. We then use the median polishing algorithm after obtaining the summarized profiles, to remove and correct for any plate position artifacts. For each feature, the algorithm de-trends the rows, i.e. by subtracting the row median from the corresponding feature of each profile in that particular row. Next, it de-trends the columns in a similar way using column medians. The row and column de-trending is repeated until convergence is reached in all the features. For the rest of the analysis we considered only the constructs which have more than 99% sequence identity to both the intended protein and gene transcript, to avoid testing uncharacterized mutations/truncations.

Not all of the morphological features contain useful reproducible information. We first filter out features for which their replicate correlation across all samples (except the negative controls) is less than 0.30, retaining 2200 features. Subsequently, a feature selection method is used (*Fischer et al., 2015*). Briefly, starting with features (measurements) that we identify as essential, a new feature that contributes the most information with respect to those that have been chosen, is added to the set. The contribution of each feature to the already-selected features is measured by the replicate correlation of the residue when the feature is regressed on the already selected features. This is repeated until the incremental information added drops below a threshold. The original method proposed in (*Fischer et al., 2015*) overfits in its regression step when the original data is very high dimensional. As a remedy, in the regression step we only use features that have a Pearson correlation of more than 0.50 with the selected features thus far. This prevents overfitting of regression when the dimensionality of selected features grows. We stop feature selection when the maximum replicate correlation of residue is less than 0.30.

The feature selection method greatly removes redundancy, but because of the non-optimal 'greedy' strategy, some redundancy remains. Principal component analysis is then applied to keep 99% of variance in data, resulting in 158 principal components being selected.

## Feature interpretation

The features measured using CellProfiler follow a standard naming convention. Each feature name is made up of several tokens separated by underscores, in the following order:

- Prefix which could be either empty or 'mad'. This means that the feature is calculated either by taking median (no prefix) or median absolute deviation ('mad' prefix) of the relevant measurement over all the cells in a sample.
- Cellular compartment in which the measurement related to the feature is made, i.e., 'Cells', 'Cytoplasm', or 'Nuclei'. Note that features labeled 'Nuclei' are based on segmentation of nuclei using Hoechst staining, 'Cells' are based on segmentation of the cell edges using the RNA channel, and 'Cytoplasm' is the subtraction of the aforementioned compartments.
- Measurement type, which can be either 'Intensity', 'Texture', 'RadialDistribution', 'AreaShape', 'Correlation_Correlation', 'Granularity', and 'Neighbors'. Note that 'Correlation_Correlation' measures, within a cellular compartment, the correlation between gray level intensities of corresponding pixel pairs across two channels (specified in the next tokens in the feature name). Note also that the relative positioning of a cell is measured in the 'Neighbors' category.
- Name(s) of channels in which the measurement is made, if appropriate (omitted for AreaShape and Neighbors).
- Feature name. The precise measurement name appears at the end. A description of each metric can be found in the CellProfiler manual (http://cellprofiler.org/manuals/current/)

## Identifying ORF constructs that are distinguishable from negative controls

Our method to identify which genes produce a discernable profile involves first normalizing each profile to the negative controls, such that a treatment's median replicate correlation becomes a surrogate for phenotype strength. In the case that a treatment does not show a phenotype different from the negative control, its replicates would center around the origin in the feature space. This would consequently decrease the median replicate correlation. On the other hand, a phenotype which is consistently observed in the replicates and is significantly different from the controls results

in the replicates to concentrate in a region far from the origin in the feature space, and hence a high median replicate correlation value.

The cutoff for 'discernible' is set based on the top fifth percentile of a null distribution. The null distribution is defined based on the correlations between non-replicates (that is, different constructs) in the experiment. Treatments whose replicate correlations are greater than the 95th percentile of the null distribution are considered as 'hits' that have a morphological phenotype that is highly reproducible (*Figure 2A*).

At this point, for strong treatments, all profiles of the replicates are collapsed by taking the average of individual features. 110 out of the 112 selected ORFs were significantly different from the untreated profiles in the feature space. That is, their average Euclidean distances to the untreated profiles were higher than 95th percentile of untreated profile distances to themselves. This shows these two alternative notions of phenotype strength–replicate reproducibility and distance to negative control–are consistent. We restrict all the remaining analyses to the 110 ORFs.

## Comparison of morphological connections between genes to protein-protein interaction data and pathway annotations

In this analysis, mutant alleles were removed and we considered only one wild-type allele for each gene with a detectable phenotype, retaining 73 genes. We calculated a threshold to identify significantly correlated gene pairs. We picked the threshold to minimize the probability of error in classifying wild-type clone pairs versus different-gene pairs. To do so, we found the value at which the probability density functions of the two groups intersect; this value (here, 0.43) can be proved to have the desired property (*Duda et al., 2012*). This approach results in about 5% of the gene pairs being categorized as highly correlated. We next formed a two by two contingency table, where the rows correspond to two groups of gene pairs, determined by whether they have high profile correlation or not. Similarly, the columns also correspond to two groups of gene pairs, determined by whether the corresponding proteins have been reported to interact in BioGRID (or alternatively have been annotated to be in the same pathway; *Supplementary file 1C and 1D*). This table was then used to perform a one-tailed Fisher's exact test.

## Creation of a dendrogram relating genes to each other, and agglomerative clustering by cutting the dendrogram

A dendrogram was created based on the Pearson correlation distance and average linkage, using the hclust function in R (*Figure 3*).

Gene clusters were formed by cutting the dendrogram at a fixed correlation level, 0.522, which was chosen using a stability-based measure. The measure is defined as follows: the local clustering stability is measured for a range of candidate cutoffs, from 0.43 (used earlier to test consistency to protein interaction data) to 0.70. The point with highest stability was chosen (*Figure 3—figure supplement 2*), and the stability measure was defined as the proportion of treatments whose clusters do not change if the cutoff is slightly changed by a small amount, $\epsilon = .002$.

## Subpopulation extraction

In order to extract cell categories (subpopulations) and subpopulation enrichment laid over the dendrogram in *Figure 3*, we applied k-means clustering on the normalized single cell data for each gene cluster and the control. Data normalization was carried out on a plate-wise basis by z-scoring each feature using the control samples as reference. In order to avoid curse of dimensionality, we restricted the dataset to the features obtained from the feature selection step mentioned earlier. We set k = 20 to be the number of subpopulations. The algorithm was run for at most 5000 iterations. Each cell was assigned to the subpopulation for which it has the shortest Euclidean distance to its center. Then, the number of cells belonging to each cell subpopulation was counted and the proportion in each subpopulation for genes in the cluster was compared against that of the control. If the change in proportion of a cell category was consistent across the genes in the cluster, the cell category is shown in the Supplementary file 2 (type B PDFs). To quantify this consistency, we used the inverse coefficient of variation of the change in a category proportion. If this quantity exceeded one, we called the change consistent and included the corresponding cell category in the PDFs.

Images of cells which have highest similarity to the category center in the feature space are then used to interpret and give name to each cell category (*Figure 3—figure supplement 3*)

## Identifying targets of a gene using a data-driven approach

For this purpose, we used a replicate of the original experiment but with L1000 gene-expression readouts, which is provided in the supplemental data; i.e. cell line, time point, and ORF constructs are the same. This data is different from the data used in creating GSEA plots, which entails multiple cell lines and time points. The mRNA levels are all normalized with respect to the negative control. For each replicate of the overexpression construct, we sort the expression levels of landmark genes and take the list of top and bottom 50 landmark genes. Then, to find targets of the gene related to the construct, we find the landmark genes among this list which has shown up at least in p% of replicates/clones of the gene. In particular, we set p to 33% for YAP1, 50% for WWTR1, TRAF2, and REL. Then, we simply take the intersection of predicted targets of YAP1 and WWTR1 (and similarly TRAF2 and REL, separately) to get their common targets. These targets are then used to produce *Figure 6—figure supplements 1–2*.

## Gene set enrichment analysis

In order to produce *Figure 6C*, we specified the three known targets of YAP/WWTR1 (CYR61, CTGF, and BIRC5) and queried for ORFs resulting in down-regulation of these genes. This scores each ORF (out of the 430 in the dataset) based on the observed change in mRNA level of the specified YAP/WWTR1 targets, across between four to nine different cell lines and between one to four time points. For each ORF, we then sought the summarized score which takes the mean of 4 largest scores across time point/cell line combinations. Finally, the ORFs were sorted based on the summarized score, and top 30 ORFs were tested for enrichment in different pathways (*Supplementary file 1I*). We used the 'clusterProfiler' package in R and the KEGG pathway enrichment analysis implemented in it for creating the GSEA plot (*Yu et al., 2012*).

## Luciferase reporter assay

Wild-type and mutant sequences of WWTR1 (TAZ) (4SA: S66A, S89A, S117A, and S311A) and YAP1 (5SA: S61A, S109A, S127A, S164A, and S397A) were previously generated and cloned into the pCMV5 backbone; these constructs are distinct from those used in the original Cell Painting data set. TRAF2 and REL were cloned from the original constructs (using Broad ID# ccsbBroadEn_01710 and ID# ccsbBroadEn_11094, respectively), into pCMV5 expression vectors. These were sequenced and confirmed to BLAST against the appropriate Broad clone ID. The empty pCMV5 backbone was used as the control condition. The Tead luciferase reporter construct, 8xGTIIC-luciferase was a gift from Stefano Piccolo (Addgene plasmid # 34615).

HEK293T cells, RRID:CVCL_0063, were transfected using Turbofect (ThermoFisher Scientific) according to manufacturer's protocol. All cells were co-transfected with a $\beta$-galactosidase reporter plasmid (pCMV-LacZ from Clontech) as a transfection control. Cells were lysed 48 hr following transfection. Lysates were mixed with firefly luciferase (Promega) according to the manufacturer's protocol and luminescence was measured using a luminometer (BioTek). Lysates were mixed with o-nitrophenyl-$\beta$-D-galactoside (ONPG) and $\beta$-galactosidase expression was determined spectrophotometrically by measurement of absorbance at 405 nm following ONPG cleavage. All luciferase readings were normalized to $\beta$-galactosidase expression for the sample. Statistical analysis was conducted using a two tailed unpaired Student's t test. The data shown in *Figure 6D* are from triplicate samples within a single experiment and is representative of replicate experiments.

## Acknowledgements

The authors gratefully acknowledge contributions from members of the Carpenter laboratory, especially Steven A Moore. We further acknowledge contributions from Federica Piccioni and Mukta Bagul in the Genetic Perturbation Platform (GPP) at the Broad Institute for helping us plan and execute the experiment. Funding for this work was provided by the National Science Foundation (NSF CAREER DBI 1148823 to AEC), a BroadNext10 grant from the Broad Institute, and the Slim Initiative for Genomic Medicine, a project funded by the Carlos Slim Foundation in Mexico.

## Additional information

### Funding

| Funder | Grant reference number | Author |
|--------|------------------------|--------|
| National Science Foundation | NSF CAREER DBI 1148823 | Anne E Carpenter |
| Eli and Edythe Broad Foundation | | Anne E Carpenter |
| Carlos Slim Foundation | | Anne E Carpenter |

The funders had no role in study design, data collection and interpretation, or the decision to submit the work for publication.

### Author contributions

MHR, SS, Conceptualization, Data curation, Software, Formal analysis, Validation, Investigation, Visualization, Methodology, Writing—original draft, Writing—review and editing; XW, Conceptualization, Data curation, Formal analysis, Investigation, Methodology, Writing—original draft, Writing—review and editing; JBB, Conceptualization, Data curation, Formal analysis, Validation, Investigation, Visualization, Methodology, Writing—original draft, Writing—review and editing; M-AB, Data curation, Software, Formal analysis, Validation, Methodology, Writing—original draft, Writing—review and editing; YS, Conceptualization, Data curation, Formal analysis, Investigation, Methodology; XV, Conceptualization, Resources, Data curation, Formal analysis, Supervision, Validation, Investigation, Visualization, Methodology, Writing—original draft, Writing—review and editing; JSB, Conceptualization, Resources, Formal analysis, Supervision, Validation, Investigation, Methodology, Writing—original draft, Writing—review and editing; AEC, Conceptualization, Resources, Formal analysis, Supervision, Funding acquisition, Validation, Investigation, Visualization, Methodology, Writing—original draft, Project administration, Writing—review and editing

### Author ORCIDs

Mohammad Hossein Rohban, http://orcid.org/0000-0001-6589-850X

Shantanu Singh, http://orcid.org/0000-0003-3150-3025

Julia B Berthet, http://orcid.org/0000-0002-6549-8116

Mark-Anthony Bray, http://orcid.org/0000-0002-9748-3592

Xaralabos Varelas, http://orcid.org/0000-0002-2882-4541

Jesse S Boehm, http://orcid.org/0000-0002-6795-6336

Anne E Carpenter, http://orcid.org/0000-0003-1555-8261

## Additional files

### Supplementary files

• Supplementary file 1. Supporting and supplemental data for the figures and experiments. (A) List of all the 323 constructs used in the experiment along with the target transcript and their public clone ID. (B) Replicate correlation is higher in the constitutively active mutant allele compared to the wild-type allele, except for AKT3_E17K. Constitutively active mutant annotations were obtained by literature search for all the mutants in the experiment showing a detectable phenotype. Genes shown here are only those where either the wild-type gene or its constitutively activating allele yielded a phenotype distinct from controls. (C) Pathways sorted based on proportion of their associated gene showing a detectable phenotype. (D) Highly correlated proteins (according to morphology in the Cell Painting assay) that have also been reported to interact physically. (E) Highly correlated genes (according to morphology in the Cell Painting assay) that have also been annotated to be related to the same pathway. (F) Gene Ontology terms associated with each gene cluster (*Alexa and Rahnenführer, 2009*). (G) Rank ordered list of distinctive features based on their z-scores for Cluster 19. (H): All genes/alleles in Cluster 8 and 10 induce cell rounding. (I) The NF-κB signaling pathway is the most enriched when searching for gene overexpressions that downregulate known YAP/TAZ targets (CYR61, CTGF, and BIRC5).

• Supplementary file 2. Type A and B PDFs are collected in a ZIP file in *Supplementary file 2*. The details of the contents have been described in *Figure 5*.

• Supplementary file 3. The CellProfiler pipeline used to process the images is released as the *Supplementary file 3*.

## Major datasets

The following dataset was generated:

| Author(s) | Year | Dataset title | Dataset URL | Database, license, and accessibility information |
|---|---|---|---|---|
| Mohammad H Rohban, Shantanu Singh, Xiaoyun Wu, Julia B Berthet, MarkAnthony Bray, Yashaswi Shrestha, Xaralabos Varelas, Jesse S Boehm, Anne E Carpenter | 2017 | Systematic morphological profiling of human gene and allele function via Cell Painting | http://idr-demo.openmicroscopy.org/webclient/?show=screen-1751 | Publicly available at the Image Data Resource (http://idr-demo.openmicroscopy.org/about/) |

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
