## [Decision Letter]

Thank you for submitting your article "Systematic morphological profiling of human gene and allele function reveals Hippo-NF-κB pathway connectivity" for consideration by *eLife*. Your article has been reviewed by three peer reviewers, and the evaluation has been overseen by a Reviewing Editor and Anna Akhmanova as the Senior Editor. The following individual involved in review of your submission has agreed to reveal their identity: Trever Bivona (Reviewer #3).

The reviewers have discussed the reviews with one another and the Reviewing Editor has drafted this decision to help you prepare a revised submission. Given the concerns about establishing a more compelling functional link between the Hippo and NF-κB pathways, we suggest that this work may be more appropriate as a Tools and Resources (TR) rather than as a Research Article (RA). In that case, it will not be necessary to perform the additional biological experiments (unless you have already performed these) and instead focus on the technical innovation of your approach. Please let us know if you wish to have the revised version considered as a TR paper. Of, course if you remain committed to having this considered as an RA, the following essential revisions will be crucial.

Summary:

The manuscript by Rohban, Carpenter et al. describes an interesting application of a technology that the Carpenter lab has established over the last few years. The potential of Cell Painting to reveal functionally meaningful phenotypes of genes of interest and the level of connection between the studied genes is the major focus on the manuscript. The library used in the analysis contains ORFs of which 220 were chosen for prioritized analysis. Of these, 50% show a notable morphological change of which 73 are unique genes. Using an unbiased clustering approach, known pathway components group together in similar phenotypes. The authors also leverage the dataset and analytical approaches to identify a novel association between the Hippo and NFKB pathways. Overall, the reviewers agree that this is a potentially useful approach to revealing novel gene functions and evaluation of the function of mutant alleles. However, a significant concern shared by all 3 reviewers is that the claimed link revealed by this analysis between the Hippo and NF-κB pathways is not adequately supported by sufficient validation and mechanistic data.

Essential revisions:

1) The report, apart from the identification of the Hippo – NFKB link is largely technical with much of the ORFs used encoding wild type alleles and thus not providing much of new insights into the biology or functional characteristics of pathogenic alleles. The link between Hippo and NFKB is substantiated by analysis of LINCS data using GSEA and with somewhat minimal additional experimental validation using reported assays in HEK293. Because this is essentially the only biological discovery of the report which is otherwise mostly technical (albeit of very high level) and is indeed claimed as a central tenant of the manuscript (see title) it needs to be further substantiated. There is insufficient confirmation of the mechanistic basis of this finding and of its relevance across non-engineered contexts, including across cancer subtypes (even just using cancer cell lines). Some insight into which genetic backgrounds are likely to be most relevant would also be helpful. The regulation of both Hippo and NFKB are complex, with many upstream signals impinging on their activity; defining some relevant context if not broadly studying which ones are and not relevant should go beyond the observation that NFKB regulation of YAP/TAZ might not be through the known upstream kinase.

2) RT-qPCR analysis should be performed to directly determine the effect of the knockdown of NF-κB pathway genes on the endogenous targets of YAP such as CTGF. Furthermore, the authors should test if NF-κB regulates the protein levels of YAP/TAZ or TEAD, their localization, their interactions with each other or with other transcription factors such as AP-1, or their binding to their transcriptional targets.

3) The exact boundaries of clusters are not always understandable to the reader. For example: STK3 has a morphological signature strongly anti-correlated to cluster 20, why is it not in cluster 11? Providing the reader with the images on which this is based in the main text would improve the readability of this manuscript.

4) What does it mean when two clusters anti-correlate? Going by the example of cell elongation, does it mean that when a certain cluster of ORFs causes cell elongation and another cluster does not cause this, they anti-correlate? Or should the opposite cluster cause cell-rounding or another morphological feature? The latter may be caused by completely different cellular processes (for example migration vs. mitosis) that should not suggest connectivity.

5) In the search for novel gene relationships the anti-correlation of cluster 11 and 20 is described. However, the authors point out that this relation is confirmed by previous work of Reginensi et al., 2013, demonstrating that YAP1 and CDC42 KO mice have a similar phenotype. This is confusing because one would expect the opposite phenotype based on this work.

6) "Confirmation" of the relation between cluster 11 and 20 is very important because this is the key biological finding in the paper (and mentioned in the title). However, only some data mining is performed on an experiment with nine different cell lines and 4 time points (time points of what?) and subsequent linking the NF-kappa-B pathway to negative regulators found in a Hippo paper. Whether or not any of those regulators is involved in the morphological phenotype described here is not addressed. Subsequently, a transcriptional reporter assay is performed with over-expression of the transcriptional regulators. This is a weak experiment based on over-expression and an artificial target gene, prone to artefacts. Without the use of knockout cell lines and a mechanistic link between the 2 pathways this part of the manuscript remains preliminary.

---

## [Author Response]

*Essential revisions:*

*1) The report, apart from the identification of the Hippo – NFKB link is largely technical with much of the ORFs used encoding wild type alleles and thus not providing much of new insights into the biology or functional characteristics of pathogenic alleles. The link between Hippo and NFKB is substantiated by analysis of LINCS data using GSEA and with somewhat minimal additional experimental validation using reported assays in HEK293. Because this is essentially the only biological discovery of the report which is otherwise mostly technical (albeit of very high level) and is indeed claimed as a central tenant of the manuscript (see title) it needs to be further substantiated. There is insufficient confirmation of the mechanistic basis of this finding and of its relevance across non-engineered contexts, including across cancer subtypes (even just using cancer cell lines). Some insight into which genetic backgrounds are likely to be most relevant would also be helpful. The regulation of both Hippo and NFKB are complex, with many upstream signals impinging on their activity; defining some relevant context if not broadly studying which ones are and not relevant should go beyond the observation that NFKB regulation of YAP/TAZ might not be through the known upstream kinase.*

*2) RT-qPCR analysis should be performed to directly determine the effect of the knockdown of NF-κB pathway genes on the endogenous targets of YAP such as CTGF. Furthermore, the authors should test if NF-κB regulates the protein levels of YAP/TAZ or TEAD, their localization, their interactions with each other or with other transcription factors such as AP-1, or their binding to their transcriptional targets.*

We thank the reviewers for their valuable feedback on the experiments needed to make the claim more rigorous. But we decided to skip further confirmatory experiments and accept the editor’s suggestion to submit the revision as a “tools” paper with an adjusted title.

*3) The exact boundaries of clusters are not always understandable to the reader. For example: STK3 has a morphological signature strongly anti-correlated to cluster 20, why is it not in cluster 11? Providing the reader with the images on which this is based in the main text would improve the readability of this manuscript.*

We note that both of the mentioned anti-correlations are around -0.50, which is moderate in value. This means it is possible that the morphological phenotype for STK3 is not highly similar to Cluster 11. Indeed, a medium anti-correlation means that only a subset of phenotypes in STK3 (and cluster 11) are opposite of that in Cluster 20. The fact that STK3 and Cluster 11 are far apart on the dendrogram means that the two subsets of opposing phenotypes are non-overlapping. We have clarified this point in the main text. Images relevant to all the clusters were provided as “cluster A” (sample image) and “cluster B” (exemplar cell images of enriched subpopulations) PDFs (Figure 5).

*4) What does it mean when two clusters anti-correlate? Going by the example of cell elongation, does it mean that when a certain cluster of ORFs causes cell elongation and another cluster does not cause this, they anti-correlate? Or should the opposite cluster cause cell-rounding or another morphological feature? The latter may be caused by completely different cellular processes (for example migration vs. mitosis) that should not suggest connectivity.*

As the reviewer suggests, anti-correlation indeed mostly results in the latter – a *change* in phenotype in the opposite direction. It does therefore follow that in some cases, such as the elongation phenotype and a mitotic “anti-phenotype”, a negative correlation may not translate to a biological meaningful connectivity*.

However, we find in many cases, and somewhat to our surprise, that a significant anti-correlation does correspond to a negative connectivity between the genes. To better understand this, let's take the subpopulation representation. Specifically, if major subpopulations of cells are oppositely enriched/de-enriched in two clusters of genes, we would observe a negative correlation between the two. In this case, treatment by members of the two clusters results in changing proportions of same subpopulations in opposite directions, which suggests a negative connectivity. We discuss this for clusters 20 and 11 in the second paragraph of the subsection “An unexpected relationship between the Hippo pathway and regulators of NF-κB signaling (Clusters 11, 20, and 22)”.

*We do note that although elongation and rounding are indeed opposite individual metrics, elongated cells and mitotic cells produce a constellation of morphological changes across the entire profile so it is unlikely in practice that mitotic cells would produce a profile anti-correlated to elongated cells.

*5) In the search for novel gene relationships the anti-correlation of cluster 11 and 20 is described. However, the authors point out that this relation is confirmed by previous work of Reginensi et al., 2013, demonstrating that YAP1 and CDC42 KO mice have a similar phenotype. This is confusing because one would expect the opposite phenotype based on this work.*

Note that sometimes over-expression of a gene results in a trans-dominant negative effect which results in precisely the opposite phenotype effect. We therefore interpret correlations and anti-correlations as connections, generally ignoring their directionality. We mention this in the Discussion section (first paragraph).

*6) "Confirmation" of the relation between cluster 11 and 20 is very important because this is the key biological finding in the paper (and mentioned in the title). However, only some data mining is performed on an experiment with nine different cell lines and 4 time points (time points of what?) and subsequent linking the NF-kappa-B pathway to negative regulators found in a Hippo paper. Whether or not any of those regulators is involved in the morphological phenotype described here is not addressed. Subsequently, a transcriptional reporter assay is performed with over-expression of the transcriptional regulators. This is a weak experiment based on over-expression and an artificial target gene, prone to artefacts. Without the use of knockout cell lines and a mechanistic link between the 2 pathways this part of the manuscript remains preliminary.*

Time point refers to the duration of treating the cells with over-expression constructs until the time gene expression readouts are made. We have clarified this in the main text.

Testing out whether the morphological phenotypes observed in YAP/TAZ and CDC42/*TRAF2* clusters are also present in the regulators found in the GSEA requires significant effort, because those regulators are not present among the genes tested in the experiment. We decided to focus our follow-up experimentation on the transcriptional regulation, which is most of interest to those who study these pathways. Hence, we carried out the reporter assays in Figure 6 in separate experiments, in addition to the data-mining exercise mentioned and shown in Figure 6.

We therefore decided to skip studying the morphological impact of the NF-κB pathway genes and accept the editor’s suggestion to submit the revision as a “tools” paper with an adjusted title.

Also, note that the regulators mentioned were obtained using an analysis based on gene expression readouts, in addition to those which were taken from the literature. Please see “Identifying targets of a gene using a data driven approach” subsection in Methods and also (Figure 6—figure supplement 1 and Figure 6—figure supplement 2).